# The Novel Benzothiazole Derivative PB11 Induces Apoptosis via the PI3K/AKT Signaling Pathway in Human Cancer Cell Lines

**DOI:** 10.3390/ijms22052718

**Published:** 2021-03-08

**Authors:** Jinsun Kim, Sung Hee Hong, So Hyun Jeon, Min Ho Park, Cha-Gyun Shin

**Affiliations:** 1Department of Systems Biotechnology, Chung-Ang University, Ansung 17546, Korea; adkan4@naver.com (J.K.); wjsthgus0309@daum.net (S.H.J.); minho089@hanmail.net (M.H.P.); 2Division of Radiation Biomedical Research, Korea Institute of Radiological and Medical Sciences, Seoul 139-706, Korea; gobrian@kcch.re.kr

**Keywords:** apoptosis, human cancer cells, novel chemical

## Abstract

Among several anti-cancer therapies, chemotherapy can be used regardless of the stage of the disease. However, development of anti-cancer agents from potential chemicals must be executed very cautiously because of several problems, such as safety, drug resistance, and continuous administration. Most chemotherapeutics selectively cause cancer cells to undergo apoptosis. In this study, we tested the effects of a novel chemical, the benzothiazole derivative N-[2-[(3,5-dimethyl-1,2-oxazol-4-yl)methylsulfanyl]-1,3-benzothiazol-6-yl]-4-oxocyclohexane-1-carboxamide (PB11) on the human cell lines U87 (glioblastoma), and HeLa (cervix cancer). It was observed that this chemical was highly cytotoxic for these cells (IC50s < 50 nM). In addition, even 40 nM PB11 induced the classical apoptotic symptoms of DNA fragmentation and nuclear condensation. The increase of caspase-3 and -9 activities also indicated an increased rate of apoptosis, which was further confirmed via Western blotting analysis of apoptosis-associated proteins. Accordingly, PB11 treatment up-regulated the cellular levels of caspase-3 and cytochrome-c, whereas it down-regulated PI3K and AKT. These results suggest that PB11 induces cytotoxicity and apoptosis in cancer cells by suppressing the PI3K/AKT signaling pathways and, thus, may serve as an anti-cancer therapeutic.

## 1. Introduction

For half a century, cancer patients have received three primary medical treatments—surgery, chemotherapy, and radiation therapy. Among these treatments, chemotherapy has the advantage of being applicable regardless of cancer stage. Over time, enormous medical progress has been made in understanding cancer biology and targeted chemotherapeutics [1,2,3,4]. New therapeutic chemicals and approaches with potent effects on tumor or healthy tissues are constantly being introduced into the clinic [5]. It is now increasingly accepted that the effectiveness of conventional chemotherapeutic drugs is in part due to their ability to induce apoptosis, although this area is not without controversy [6,7].

Benzothiazole derivatives are heterocyclic compounds with oxygen, nitrogen, and sulfur in their structures [8,9]. They have wide ranges of bioactivities, such as anti-diabetic [10], anti-microbial [11], anti-inflammatory [12], anti-fungal [13], and anti-neoplastic activities [14,15]. Recently, new diverse compounds synthesized using the rational drug designing approaches have been reported to have remarkable effects as anti-cancer drugs [16,17,18]. These compounds target the regulatory pathways of various biological events and critical factors that are essential for the survival of cancer cells. These targets include DNA replication, transcription, translation, and mitosis [19].

Recently, SH Hong and his colleagues, one of our authors, reported several potential anti-cancer compounds that have been selected from a chemical library obtained from the ChemBridge (San Diego, CA, USA) [20,21,22]. Several candidates were obtained from the library screening. Benzothiazole derivative is one of the candidates from above. In this study, a novel benzothiazole derivative, N-[2-[(3,5-dimethyl-1,2-oxazol-4-yl)methylsulfanyl]-1,3-benzothiazol-6-yl]-4-oxocyclohexane-1-carboxamide (PB11), was tested for potential use as an anticancer compound (Figure 1). By targeting the apoptosis pathway with the aim of inducing cytotoxicity, experiments were designed and conducted on human cancer cell lines U87 and HeLa.

## 2. Results and Discussion

### 2.1. PB11 Is Highly Cytotoxic at the nM Scale

First, cytotoxicity of PB11 was investigated with both non-cancer cell lines and cancer cell lines. It was shown that PB11 is less cytotoxic to non-cancer cells compared to cancer cell lines [Data not shown]. Among the cancer cell lines tested, the viabilities of U87 and HeLa cells were significantly reduced in the presence of PB11, as assessed via the 2,5-diphenyl tetrazolium bromide (MTT) and lactate dehydrogenase (LDH) assays (Figure 2). In the case of U87 cells, treatment with 10 nM, 100 nM, 1 μM, or 10 μM PB11 decreased the cell viability to 85.91 ± 1.25%, 37.25 ± 4.02%, 19.23 ± 0.78%, and 6.56 ± 0.15%, respectively, of the untreated control cells. The same concentrations of PB11 decreased the viability of HeLa cells to 86.82 ± 5.2%, 46.23 ± 2.18%, 21.46 ± 1.42%, and 11.34 ± 1.03%, respectively, of the untreated control. Based on the dose-response curves, the IC50 for PB11 was estimated to be approximately 40 nM (Figure 2A). 

Lactate dehydrogenase (LDH) is a cytosolic enzyme present in many cell types and is released into the media of cell cultures when the plasma membrane is damaged [23]. The LDH cytotoxicity assay is a colorimetric assay that provides a simple and reliable method for determining cytotoxicity. In the case of the U87 cells, the LDH cytotoxicity levels of the cells treated with 0 nM (mock), 10 nM, 100 nM, 1 μM, and 10 μM PB11 were 8.41 ± 0.35%, 24.04 ± 1.60%, 59.90 ± 0.08%, 84.20 ± 0.70%, and 93.51 ± 1.04%, respectively, of the control, which was entirely lysed untreated cells (Figure 2B). In addition, the LDH cytotoxicity levels in HeLa cells were 9.27 ± 0.10%, 19.72 ± 0.97%, 60.32 ± 1.35%, 76.68 ± 1.73%, and 93.88 ± 0.63%, respectively. As shown in Figure 2, the LDH assay results strongly supported the MTT assay results. Altogether, these results show that PB11 is highly cytotoxic to U87 and HeLa cells at the nM scale.

### 2.2. PB11 Induces Apoptosis through Mitochondria

We hypothesized that the reduced cell number observed upon PB11 treatment might be associated with apoptosis. Therefore, PB11-treated cells were assessed for the classical apoptosis indicators DNA and nuclear fragmentation.

Very distinctive DNA fragmentation was detected in U87 and HeLa cells treated with 40 nM PB11 (Figure 3A). In this experiment, we used 5 μM camptothecin, which is a well-known apoptotic drug, as a positive control [24]. PB11 induced DNA fragmentation very clearly and effectively as 40 nM PB11 induced greater DNA fragmentation than 5 μM camptothecin.

Furthermore, DAPI (4′,6-diamidino-2-phenylindole) DNA staining revealed that PB11 caused alterations in the nuclear morphologies of U87 and HeLa cells, such as nuclear condensation and fragmentation, which are suggestive of apoptosis (Figure 3B, arrow).

Caspases are recognized as the key enzymes of apoptosis [25]. To investigate whether they are associated with PB11-induced cytotoxicity, the levels of apoptotic markers in PB11-treated cells were evaluated (Figure 4A). We observed that the levels of Bax, cytochrome c, and cleaved caspase-3, which is the active form, were increased in PB11-treated cells. These data support our conclusion that PB11 induces cytotoxicity via apoptosis.

Apoptosis occurs via the intrinsic or extrinsic pathway. The intrinsic pathway, also called the mitochondrial pathway, is activated by intracellular signal in intermembrane space of mitochondria, whereas the extrinsic pathway is activated by ligand binding interaction in extracellular surface [26]. To understand which pathway is involved in the apoptotic process of PB11-treated cells, the activities of caspase-3, -8, and -9 were evaluated by using colorimetric assay (Figure 4B). PB11 treatment of the cells increased the activities of caspase-3 and -9 by time. After 24 h and 48 h treatment, the activities were increased by approximately three- and five-fold, respectively. Conversely, the caspase-8 activities in the cells were unaffected. Therefore, PB11 seems to induce cytotoxicity via the intrinsic pathway of apoptosis.

### 2.3. PB11 Induces Apoptosis via the PI3K/AKT Signaling Pathway

To gain further insight into the molecular mechanisms underlying the PB11-induced apoptosis, we decided to screen the expression levels of several signal-transducing proteins, such as PI3K, JNK, and AKT, which are directly related to cell proliferation and death. Hyper-activation of the PI3K/AKT signal-transduction pathway leads to hyperplasia and neoplastic transformation, whereas inhibition of the pathway is frequently associated with cell death [26]. The c-Jun N-terminal kinase (JNK) pathway is involved in cell growth associated with a wide range of abiotic and biotic stresses [27]. The NFκB/IκB pathway is important in regulating cellular physiological and immunological statuses [28].

Western blot analysis for members of these pathways revealed that PB11 treatment decreased the levels of phosphorylated (p-) PI3K and p-AKT, which are the active forms of PI3K and AKT, respectively (Figure 5). Conversely, the levels of active JNK (p-JNK) and NFκB (p-NFκB) were unaffected. These results indicate that PB11 represses the PI3K/AKT pathway but does not affect the JNK and NFκB pathways, and consequently, the cells undergo apoptosis.

Taken together, it is proposed that the novel benzothiazole derivative PB11 may be beneficial for the treatment of several cancers. Additionally, these results support the conclusions of a previous study wherein benzothiazole suppressed the migration and invasion of cancer cells [17,18,29]. Furthermore, one of the benzothiazole derivatives containing carboxamide also showed anti-cancer effect to the MCF-7 cell line [30]. Both PB11 and that derivative have high cell cytotoxicity in nM scale. The PI3K/AKT pathway has been shown to be aberrantly activated in several cancer types and is responsible for the emergence, growth, and development of various cancers in humans [31]. Accordingly, it is an important target for the management of cancers in humans. Interestingly, this study found that the novel benzothiazole PB11 suppresses the PI3K/AKT pathway in cancer cells, highlighting the potential of this compound as a therapeutic against various cancers.

## 3. Materials and Methods

### 3.1. Reagents

PB11 was obtained from Chembridge (San Diego, CA, USA). Each dose of PB11 was prepared by diluting with 40% dimethyl sulfoxide (DMSO). 2,5-diphenyl tetrazolium bromide (MTT) and DAPI reagents were purchased from Sigma-Aldrich (St. Louis, MO, USA). The LDH assay kit was obtained from Dongin Biotech (Seoul, Korea). Caspase-3 and -9 assay kits were purchased from Promega (Madison, WI, USA), and the caspase-8 assay kit from Biovision (Milpitas, CA, USA). The antibodies against Caspase-3, Bax, Cytochrome C, IκBα, NFκB p65, JNK, p-JNK, AKT, p-AKT, and PI3K were purchased from Santa Cruz (Santa Cruz, CA, USA). The p-PI3K monoclonal antibody was purchased from Cell Signaling Technology (Danvers, MA, USA). The rabbit anti-B-actin antibody was purchased from Thermo Fisher Scientific (Waltham, MA, USA).

### 3.2. Cell Culture

The human cancer cell lines U87 and HeLa cells (Korean Cell Line Bank, Seoul, Korea) were cultured in Dulbecco’s modified Eagle’s Medium (DMEM) supplemented with 2 mM L-glutamine, 10% heat-inactivated fetal bovine serum (Sigma-Aldrich, St. Louis, MO, USA), 100 U/mL penicillin, and 100 μg/mL streptomycin. All cultures were performed at 37 °C with 5% CO_2_.

### 3.3. Cell Viability and Toxicity Tests

#### 3.3.1. MTT Assay

The MTT assay was conducted as previously described [32,33]. Briefly, cells were inoculated in 96-well plates (50 μL of 4 × 10^4^ cells/well). After 4 h, 50 μL of fresh medium containing PB11 at the indicated doses was added to each well. After 48 h of continuous exposure, 50 μL of 0.1 mg/mL MTT solution was added into each well, and the samples were incubated at 37 °C for 4 h. After discarding the supernatants, the cells were incubated with 200 μL of 100% (*w*/*v*) DMSO at 25 °C for 10 min. The absorbance at 595 nm was measured using a microplate reader. Three independent experiments were conducted in duplicate at different time points.

#### 3.3.2. LDH Cytotoxicity Assay

The LDH assay was conducted using the D-Plus™ LDH cell cytotoxicity assay kit (Dongin Biotech, Seoul, Korea). Briefly, 2.5 × 10^4^ cells per well were seeded in 96-well plates. After 24 h of incubation, PB11 at the indicated doses was added into each well in the final volume of 100 µL. After 48 h, floating cells in the supernatants were removed using centrifugation at 600× *g* for 10 min. Control cells were lysed by adding 10 µL lysis buffer before centrifugation. Each supernatant (10 µL) was transferred to a new well in a 96-well plate. Finally, 100 µL of the LDH reaction mixture (1:50 ratio of WST substrate to LDH assay buffer) was added, and the samples were incubated at 25 °C for 30 min. The absorbance at 450 nm was measured using a microplate reader. Three independent experiments were conducted in duplicate at different time points.

### 3.4. Assessment of Apoptosis

#### 3.4.1. DNA-Fragmentation Assay

Low–molecular-weight DNA was extracted from cells as previously described [34]. Cells grown in 100 mm plates were treated with 40 nM PB11 for 48 h and then rinsed with phosphate-buffered saline (PBS) before harvested. Positive control was conducted with 5 µM camptothecin. The cells were resuspended in ice-cold lysis buffer (10 mM Tris [pH 7.5], 0.2% Triton X-100, and 10 mM EDTA) and incubated on ice for 30 min. The lysates were centrifuged at 10,000× *g* at 4 °C for 10 min, and the supernatants were consecutively extracted with buffered phenol, buffered phenol–chloroform, and chloroform–isoamyl alcohol (24: 1, vol/vol). The DNA was ethanol-precipitated and then resuspended in 10 mM Tris (pH 7.5) with 1 mM EDTA, treated with RNase A for 30 min at 37 °C, and then analyzed via electrophoresis on a 1.5% agarose gel.

#### 3.4.2. Evaluation of Nuclear Morphology

The morphological changes in the nuclei of PB11-treated cells were examined via DAPI-staining, as described previously [32]. Cells were seeded in 8-well plates (1.5 × 10^5^ cells/well) and treated under the same conditions as those used for the MTT assay described above. After 48 h incubation period, the medium was removed, and the cells were washed three times with PBS. Afterward, the cells were fixed with 4% formaldehyde with 0.1% triton X-100 for 20 min at 25 °C. Next, these fixed cells were stained with 10 mM DAPI in 1× PBS for 1 h at 37 °C. The samples were visualized using a Nikon fluorescence microscope (TE 2000 u; Tokyo, Japan) with ultraviolet (UV) excitation at the wavelengths between 300 and 500 nm.

#### 3.4.3. Caspase Activity Assay

The caspase-3, -8, and -9 activities were measured using the caspase-3, -8, and -9 colorimetric assay kits (Promega, Biovision, USA), respectively [32,34]. U87 and HeLa cells were treated with 40 nM PB11 for 0, 24, or 48 h. Then, 50 μL of the cell-lysis buffer was added to the cells, and the samples were incubated for 10 min on ice. Subsequently, the samples were centrifuged at 10,000× *g* for 1 min, and the total protein concentration of each supernatant was quantified by Bradford assay. Afterward, each lysate of 20 μg total protein was mixed with 50 μL of 2× reaction buffer and 4 mM DEVD-pNA substrate or 4 mM LEHD-pNA substrate from assay kits. After incubating for 1 h at 37 °C, the absorbance of the samples at 405 nm was measured using a spectrophotometer.

### 3.5. Western Blotting Analysis

Cells were seeded in 6-well plates (2.0 × 10^5^ cells/well). After 24 h, the cells were treated with 40 nM PB11. Untreated and treated cells were lysed in RIPA buffer (50 Mm Tris-HCl [pH 7.4], 0.1% SDS, 0.5% sodium deoxycholate, and 150 mM NaCl). The lysates were centrifuged at 20,000× *g* for 15 min at 4 °C. Total-protein concentration was measured using the Bradford assay. Blotting was conducted as previously reported [33]. Briefly, the proteins of each lysate (equivalent of 10 μg total protein) were resolved via SDS-PAGE at 130 V for 1.5 h. The resolved proteins were then transferred onto nitrocellulose membranes (GE Healthcare UK Ltd., Hammersmith, UK) at 40 V for 1.5 h via a semi-dry–transfer apparatus (Hoefer, Inc., Holliston, MA, USA). The membranes were blocked for 3 h at 25 °C with the blocking buffer PBST (5% [*w*/*v*] non-fat dry milk and 0.1% [*w*/*v*] Tween 20 in PBS]. Finally, the membranes were then probed with the appropriate monoclonal antibodies against apoptosis-associated proteins (1:1000 dilution) in PBST solution for 1 h. After washing with PBST, the membranes were incubated with goat anti-rabbit IgG conjugated to horseradish peroxidase (1:10,000 dilution, Sigma-Aldrich, USA) or goat anti-mouse IgG conjugated to horseradish peroxidase (1:5000 dilution, Abcam, Cambridge, UK) in PBST for 1 h at room temperature. The membranes were washed three times with PBST and developed with a chemiluminescence detection kit (BioFACT, Daejeon, Korea). As an internal control, β-actin was probed with a mouse monoclonal antibody (1:5000 dilution, Thermo Fisher Scientific, Waltham, MA, USA).

### 3.6. Statistical Analysis

All data are expressed as mean ± SEM. Statistical significance was analyzed with the two-paired Student’s *t*-test; * = *p* < 0.05, ** = *p* < 0.01, and *** = *p* < 0.001.

## 4. Conclusions

In this study, the results suggest that PB11 is highly cytotoxic to cancer cells because it induces apoptosis by suppressing the PI3K/AKT signaling pathway.

## Figures and Tables

**Figure 1 ijms-22-02718-f001:**
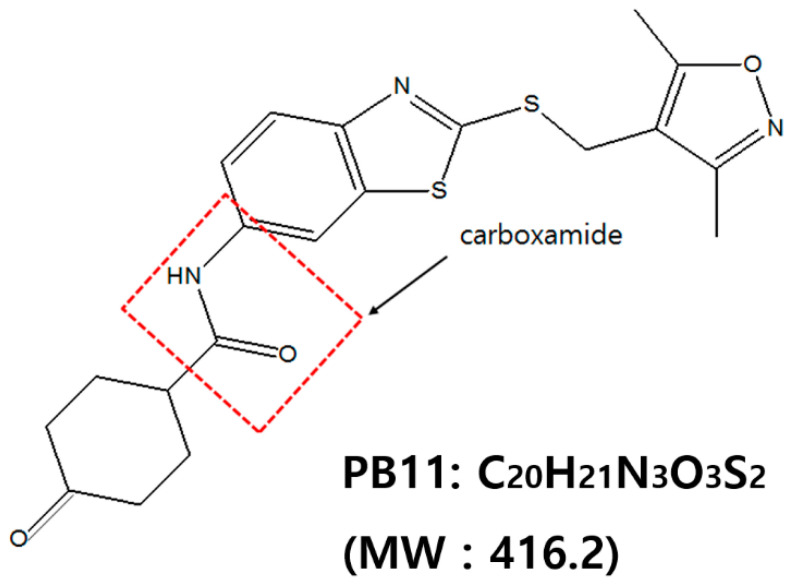
Chemical structure of N-[2-[(3,5-dimethyl-1,2-oxazol-4-yl)methylsulfanyl]-1,3-benzothiazol-6-yl]-4-oxocyclohexane-1-carboxamide (PB11, MW: 416.2).

**Figure 2 ijms-22-02718-f002:**
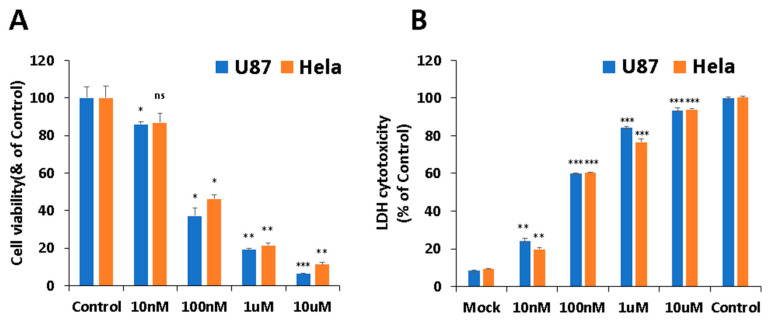
Cell cytotoxicity of PB11 in cancer cells. U87 and HeLa cells were grown in 96-well culture plates with 0 nM, 10 nM, 100 nM, 1 μM, or 10 μM PB11. After 48 h, the cytotoxicity of PB11 was evaluated by determining the number of viable cells via the 2,5-diphenyl tetrazolium bromide (MTT) assay (**A**) or by determining the amount of lactate dehydrogenase (LDH) released into culture (**B**). The cell viability and released LDH amount are presented as percentages of the control (mock-treated cells in (**A**) and lysed untreated cells in (**B**)). Based on the dose-response curves, the IC50 for PB11 was estimated to be approximately 40 nM. All data are expressed as mean ± SEM. Statistical significance was analyzed with the two-paired Student’s *t*-test; ns = *p* > 0.05, * = *p* < 0.05, ** = *p* < 0.01, and *** = *p* < 0.001.

**Figure 3 ijms-22-02718-f003:**
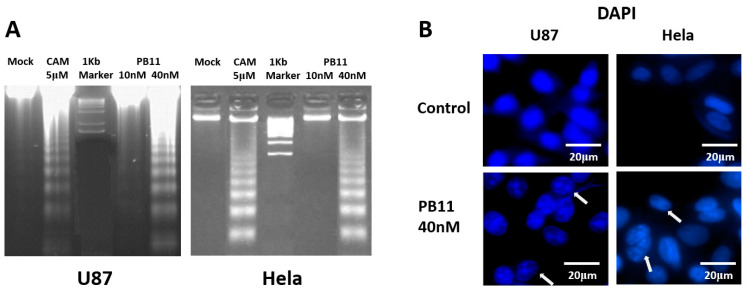
Apoptotic fragmentation of PB11-treated cancer cells. (**A**) DNA fragmentation. U87 and HeLa cells were grown in 6-well culture plates with 10 or 40 nM PB11. After 48 h, the cells were harvested and lysed. DNA was precipitated by ethanol. As a positive control for DNA fragmentation, cells treated with 5 μM camptothecin (CAM) were likewise investigated. (**B**) Nuclear fragmentation and condensation. Both cells were grown in 8-well culture plates with 40 nM PB11. The cell nucleus was stained using DAPI. Scale bar is denoted in figure.

**Figure 4 ijms-22-02718-f004:**
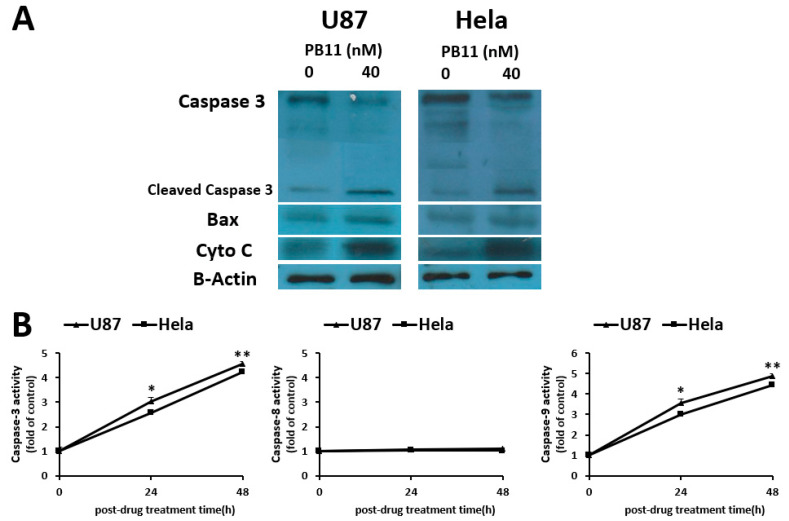
The increased levels of apoptosis markers in PB11-treated cells. (**A**): Western blot of apoptotic proteins. U87 and HeLa cells were grown in 6-well culture plates with 40 nM PB11. After 48 h, cell lysates were prepared and blotted with antibodies against caspase-3, Bax, and cytochrome C. β-actin was used as loading control. (**B**): Comparison of caspase-3, -8, and -9 activities. Cells were treated with 40 nM PB11. Cell lysates were prepared after 24 h and 48 h. The caspase activities were assessed using commercial kits. All data are expressed as mean ± SEM. Statistical significance was analyzed with the two-paired Student’s *t*-test; ns = *p* > 0.05, * = *p* < 0.05, ** = *p* < 0.01.

**Figure 5 ijms-22-02718-f005:**
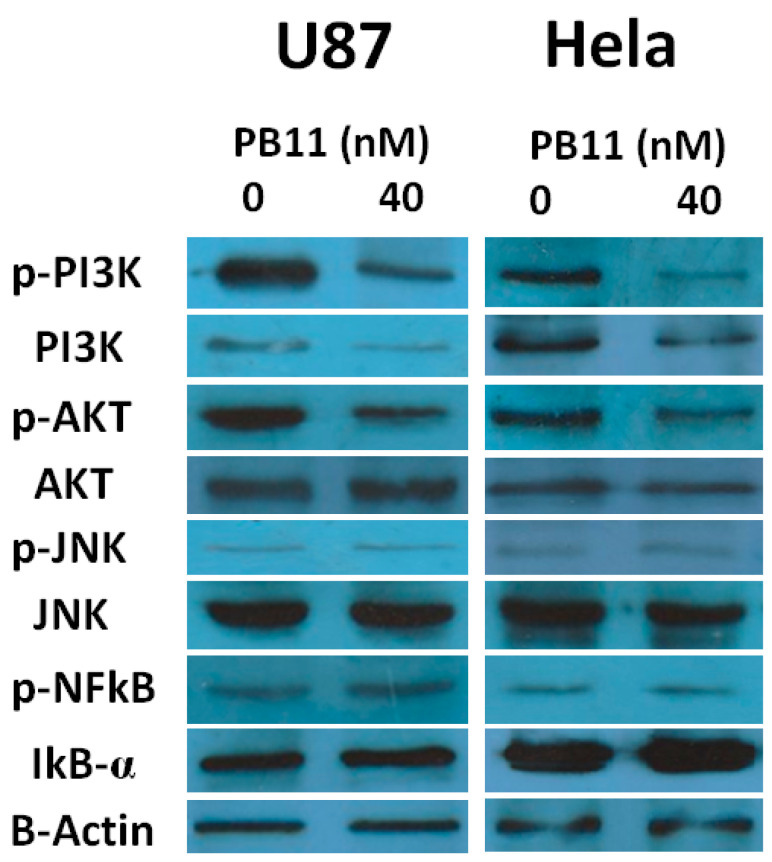
Suppression of the PI3K/AKT pathway by PB11. U87 and HeLa cells were grown in 6-well culture plates with or without 40 nM PB11. After 48 h, cell lysates were prepared and blotted with antibodies against PI3K, p-PI3K, JNK, p-JNK, IκBα, NFκB p65, AKT, and p-AKT. B-actin was used as a loading control.

## Data Availability

The data that support the findings of this study are available from the corresponding author upon reasonable request.

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
