# Peer review of "The Novel Benzothiazole Derivative PB11 Induces Apoptosis via the PI3K/AKT Signaling Pathway in Human Cancer Cell Lines"

_ijms, 2021, doi:10.3390/ijms22052718_

Round 1

Reviewer 1 Report

The authors investigated the effects of a novel benzothiazole derivative PB11 on proliferation and release of LDH in U87 and HeLa cancer cell lines. They showed that PB11 increases DNA fragmentation similar to induction of apoptosis. Western blot results suggest that PB11 inhibits phospho-AKT but has no effects on JNK and NFkB pathways.

This study provides a minimal advance on whether PB11 can be used as anti-cancer agent; no in vivo animal experiments are provided.  No information of PB11 effect on non-cancer cells is provided, thus the study does not address whether PB11 is toxic to all cells. Whether PB11 inhibition of P-AKT is the main mechanism of its effect needs further investigation.  This could be a secondary effect not relevant to its functional effect. No information as to potential target or targets for PB11 is mentioned.  Or why PB11 is a better option compared to other benzothiazole compounds.  Fig. 4a, cytochrome c release from the mitochondria is considered a marker for apoptosis.  Relevance of total cyto c increase not clear.  Use of caspase inhibitor such as QVD (10 uM) to determine if it blocks PB11 effect would be helpful.

Author Response

We appreciate your invaluable comments and advice on our manuscript.

Regarding to English and style, our manuscript was previously reviewed by a professional commercial editing group, Wordvice. A certificate for English editing is included as a separate file. If there is still wrong expression in the manuscript, we will further revise the manuscript if you pinpoint.

Regarding to Research design and methods, we have studied a new chemical for its cytotoxicity and potential use as anti-cancer agent by using the in vitro cell culture system. I think many researchers worked in this way when they studied new chemicals to investigate its effect on cells before going to animal experiments.

Regarding to Research result and conclusion, it is very difficult to make nice results and good conclusion from raw date. As results of our work, we just described the facts derived from the raw date which we had. In addition conclusion was made from the results without exaggeration. We know more works are needed for exact evaluation of the compound, PB11. However, it is just a suggestion for possibility of PB11 as an anti-cancer agent. I appreciate your understanding.     

Question 1. Effect of PB11 on non-cancer cell?

Answer: Previously we did cytotoxicity test on non-cancer cell as well as cancer cell line. Relatively non-cancer cell was less sensitive to PB11, compared to cancer cells. The date is attached as a separate file.       Line 61-63

Question 2. More detailed investigation is needed for PB11 inhibition of pAKT ?

Answer: We absolutely agree with Reviewer’s suggestion. We are planning to study with other signal transduction components which are related to AKT signal transduction pathway. But to do that, it will take long time.

Question 3. Is PB11 better than other benzothiazole compounds?

Answer: As it was described in the introduction part of the manuscript, they are very diverse in structure and show various biological activities. Some compounds showed anti-neoplastic activities. Really it is hard to say that PB11 is better than other derivatives because their activities will be different depending on the target cell or tissue. One of the derivative was introduced to have anti-cancer activity in the Result and Discussion section.     Line 147-149

Question 4. In Fig 4a, use a caspase inhibitor such as QVD (10 uM) to see if it block apoptosis caused by PB11.

Answer: We did MTT experiment with Z-VAD-FMK, a caspase inhibitor. In this experiment viability of the cells treated with Z-VAD-FMK and PB11 is much higher than that of the cells treated with only PB11. These date are not included in the manuscript, but now it will be attached as a separated file. Therefore we did not checked caspase3 activity in the presence of caspase inhibitors. However we will do that experiment if you think that the result should be included. Previously we did same assay in our past publication (reference 34).  

Reviewer 2 Report

In this paper, authors describe the cytotoxic activity of benzothiazole derivative PB11 on human cancer cell lines (U87 and HeLa). Generally speaking this paper based on authors patent (“Preparation of benzothiazole derivatives useful in treatment of cancer” US20160016943). The PB11 have good activity against tumor cell lines studied and paper could be of general interest, but two major drawback should be overcome before the publication:

  • It is absolutely not clear how PB11 was designed? Blind screening? Rational e/o computer aided design? This issue should be clarified.
  • The cytotoxic activity of PB11 on tumour cell lines alone is not enough to make it good antitumor drug candidate. The therapeutic index should be calculated, conducting the experiments on non-transformed cells.

Minor issue: the molecular formula and chemical name of PB11 in Figure 1 are not correspond to it chemical structure.

Author Response

We appreciate your invaluable comments and advice on our manuscript.

Regarding to English and style, our manuscript was previously reviewed by a professional commercial editing group, Wordvice. A certificate for English editing is included as a separate file. If there is still wrong expression in the manuscript, we will further revise the manuscript if you pinpoint.

Regarding to Research design, we have studied a new chemical for its cytotoxicity and potential use as anti-cancer agent by using the in vitro cell culture system. I think many researchers worked in this way when they studied new chemicals to investigate its effect on cells before going to animal experiments.

Regarding to Research result and conclusion, it is very difficult to make nice results and good conclusion from raw date. As results of our work, we just described the facts derived from the raw date which we had. In addition conclusion was made from the results without exaggeration. We know more works are needed for exact evaluation of the compound, PB11. However, it is just a suggestion for possibility of PB11 as an anti-cancer agent. I appreciate your understanding.     

Question 1. Why is PB11 selected for research?

Answer: One of our author (Sung Hee Hong) bought a chemical library from the ChemBridge (San Diego, CA, USA) and screened inhibitor for peroxisome proliferator-activated receptor (PPAR, a potential target for anti-cancer). Then he and his lab members published three article last year. PB11 is one of the compound screened at that time. We collaborate to study biological effect of PB11 on various cells. The reference are added on the text.   Line 44-48   

Question 2. Wrong chemical name and molecular formula of PB11.

Answer: We are very sorry for this mistake. We really appreciate your indication. The chemical name and formula were revised after consulting with a chemist.                               Line 48-51 and Fig 1 legend     

Round 2

Reviewer 1 Report

I disagree that many researchers provide a minimal in vitro analysis of new chemicals for publication before proceeding to in vivo study. As per author's response "It is very difficult to make nice results and good conclusion from raw date. However, it is just a suggestion for possibility of PB11 as an anti-cancer agent." This suggests data is premature for publication. Also, title indicates role via PI3K/AKT pathway, and the data provided is not convincing. At the very least, the two supplemental figures stated that will not be included in paper, should be included to provide support. My previous concern in Fig. 4a (cytochrome c release from the mitochondria is considered a marker for apoptosis.  Relevance of total cyto c increase not clear) was not addressed. This is important because the difference is greater than the rather modest decrease in P-AKT. Overall, there is low enthusiasm for this paper.  

Reviewer 2 Report

Authors state: "First, cytotoxicity of PB11 were investigated with both non-cancer cell lines and cancer cell lines. It was showed that PB11 is less cytotoxic to non-cancer cells, compared to cancer cell lines". In their experiment they compare the toxicity on 293T cells with glioblastoma and cervical cancer cells. Probably the better model of non-transformed cells could be chosen, but it is better than nothing. I suggest to place the toxicity diagram into the main text adding the statistical analysis.